# Is the Glycoprotein Responsible for the Differences in Dispersal Rates between Lettuce Necrotic Yellows Virus Subgroups?

**DOI:** 10.3390/v14071574

**Published:** 2022-07-20

**Authors:** Eko Y. Prabowo, Gardette R. Valmonte-Cortes, Toni Louise Darling, Elizabeth Buckley, Mark Duxbury, Brent Seale, Colleen M. Higgins

**Affiliations:** 1School of Science, Auckland University of Technology, Auckland 1010, New Zealand; ekoyakso.prabowo@pom.go.id (E.Y.P.); gardette.valmonte@aut.ac.nz (G.R.V.-C.); darlingtoni@hotmail.com (T.L.D.); elizabeth.buckley@aut.ac.nz (E.B.); mark.duxbury@aut.ac.nz (M.D.); brent.seale@aut.ac.nz (B.S.); 2Department of Microbiology and Molecular Biology Testing of Drugs and Food, The National Agency of Drug and Food Control, Jakarta 10560, Indonesia

**Keywords:** lettuce necrotic yellows virus, LNYV, glycoprotein, rhabdovirus, cytorhabdovirus, plant virus, virus–insect interaction

## Abstract

*Lettuce necrotic yellows virus* is a type of species in the *Cytorhabdovirus* genus and appears to be endemic to Australia and Aotearoa New Zealand (NZ). The population of lettuce necrotic yellows virus (LNYV) is made up of two subgroups, SI and SII. Previous studies demonstrated that SII appears to be outcompeting SI and suggested that SII may have greater vector transmission efficiency and/or higher replication rate in its host plant or insect vector. Rhabdovirus glycoproteins are important for virus–insect interactions. Here, we present an analysis of LNYV glycoprotein sequences to identify key features and variations that may cause SII to interact with its aphid vector with greater efficiency than SI. Phylogenetic analysis of glycoprotein sequences from NZ isolates confirmed the existence of two subgroups within the NZ LNYV population, while predicted 3D structures revealed the LNYV glycoproteins have domain architectures similar to Vesicular Stomatitis Virus (VSV). Importantly, changing amino acids at positions 244 and 247 of the post-fusion form of the LNYV glycoprotein altered the predicted structure of Domain III, glycosylation at N248 and the overall stability of the protein. These data support the glycoprotein as having a role in the population differences of LNYV observed between Australia and New Zealand.

## 1. Introduction

*Lettuce necrotic yellows virus* is a species in the genus *Cytorhabdovirus* and appears to be endemic to Australia and Aotearoa New Zealand (NZ) [1]. Plant hosts of lettuce necrotic yellows virus (LNYV) include *Lactuca sativa* (lettuce), the weed *Sonchus oleraceus* (sowthistle) and the model species *Nicotiana glutinosa*. The virus is vectored in a persistent, circulative and propagative manner by the aphids *Hyperomyzus lactucae* (blackcurrant-sowthistle aphid [2,3], *H. carduellinus* (Asian sowthistle aphid) [3], and *Nanosovia ribisnigri* (blackcurrant lettuce aphid) [4]. LNYV causes a severe disease in lettuce, which results from transmission by the aphid vectors as they probe for a suitable host to colonise—lettuce is not a natural food with aphids having been shown to eat it only under starvation conditions in the laboratory [1,5]. Nevertheless, increasing numbers of aphids have been correlated with increasing numbers of cases of LNYV infection in lettuce [6]. While disease management strategies through weed and insect management are advised, LNYV remains a threat to the lettuce industry in both Australia and NZ [4,7,8].

LNYV is monopartite, with a single-stranded, negative-sense RNA genome of about 12.8 kb with 3′ leader and 5′ trailer sequences that show complementarity [9]. The genome encodes six proteins, namely: a nucleocapsid (N), a phosphoprotein (P), a movement protein (4b), a matrix protein (M), a glycoprotein (G) and a polymerase (L). The virion is enveloped, and bacilliform, with a genome coated with N proteins is bound to P and L proteins. This complex is surrounded by M proteins that link the G proteins to the N proteins. The layer of M proteins is surrounded by a lipid envelope, through which the G proteins protrude. These G proteins cover the virus’s outer surface [1].

While less is known about the cytorhabdovirus/aphid interaction, it has been suggested that the glycoprotein is essential for virion attachment and penetration of insect host cells [10,11] and may serve as a model for how LNYV interacts with its aphid hosts. When probing an infected plant, an aphid may pick up virions into its stylet from where it enters the digestive tract, the midgut, the haemolymph and into the salivary gland, replicating in the latter two tissues. Once in the salivary gland, the virus can be delivered to a new host plant through the stylet of the probing insect [12]. Although digestive enzymes and the insect’s innate immune response could prevent the virus’s translocation in the midgut, the rhabdovirus could overcome these through receptor-mediated endocytosis, mediated by its glycoprotein [10,13,14,15]. Consequently, the glycoprotein is likely to play a vital role for LNYV infecting the insect host, and variation within it may impact the outcome of the virus/host interaction.

Studies on the viral nucleocapsid (N) nucleotide and amino acid sequences show the LNYV population is made up of two subgroups, SI and SII [16,17]. The LNYV population in Australia appears to be dominated by SII, with SI not detected since 1993 and, therefore, apparently extinct [16]. While the LNYV population of NZ needs to be explored more fully, it comprises both subgroups, with subgroup I predominating [17]. It has been hypothesised that SII has become dominant in Australia through more rapid dispersal than SI, with one possibility being due to more efficient transmission of SII by the insect host [17]. Ongoing analysis of the LNYV population in both countries is of value, as is the investigation of possible mechanisms for changes in population structure. To this end, this study used phylogenetics and predicted 3D protein structures of the LNYV-SI and -SII glycoprotein to determine if protein variations could explain the observed viral population structures.

## 2. Materials and Methods

### 2.1. Virus Samples

Lettuce leaves symptomatic of LNYV infection were collected from various locations in Aotearoa New Zealand (Figure 1A). Isolates HV14 and HV18 were provided by Colleen Higgins (Auckland University of Technology, Auckland, New Zealand), while the remaining isolates were provided by John Fletcher (The Institute for Plant and Food Research, NZ, Figure 1 and Table 1). RNA was extracted from these leaves using a Spectrum Plant Total RNA Kit (Sigma-Aldrich, St. Louis, MO, USA). Subgroup identification for each isolate based on the nucleocapsid gene sequence had been done previously (Figure 1A, [18]).

### 2.2. Analysis of the Glycoprotein Gene Sequences

Primer3 [19,20,21] within Geneious R6 v6.1.8, Biomatters Ltd., Auckland, New Zealand (https://www.geneious.com, accessed on 5 October 2020) was used to design primers for the amplification and sequencing of the glycoprotein gene from each subgroup (Appendix A). These were designed based on unpublished sequences of LNYV-NZ SI and SII genomes and the published LNYV-SI genome from Australia (NC_007642). Primers P5 and P6, and P1 and P2, were used to amplify the whole G gene from SI and SII, respectively, using the Invitrogen SuperScript™ IV Reverse-Transcriptase Polymerase Chain Reaction (RT-PCR) System (Invitrogen, Carlsbad, CA, USA), with the following components: 400 ng of RNA template, 500 nM of forward and reverse primers, 12.5 μL of 2X reaction mix, 0.25 μL of SuperScript™ IV RT mix and nuclease-free water to a final volume of 25 μL. The RT-PCR conditions were: reverse transcription for 10 min at 55 °C, RT inactivation/initial denaturation for two minutes at 98 °C, amplification for 40 cycles with denaturation for 10 s at 98 °C, annealing for 10 s at 50 °C and extension for one minute at 72 °C, then one cycle of final extension for five minutes at 72 °C. PCR products of the expected sizes (Appendix A) were purified from 1% agarose/1XTBE gels containing 0.5 μL Redsafe (iNtRON Biotechnology, Sangdaewon-dong, South Korea) using the GeneJET Gel Extraction Kit (Thermo Fisher Scientific, Carlsbad, CA, USA) with an elution volume of 30 μL. Purified PCR products were Sanger sequenced by Macrogen Inc. (Seoul, South Korea) and sequenced using the subgroup-appropriate G Gene ORF primers and internal primers (Appendix A). Sequences were curated and assembled using Geneious R6.

Multiple sequence alignments were carried out with nucleotide (nt) and amino acid (aa) sequences of each isolate, LNYV-SI AU2 (NC_007642) [22], lettuce yellow mottle virus (LYMoV, EF687738) [23] and Sonchus yellow net nucleorhabdovirus (M73626) using MUSCLE [24] in Geneious v6.1.8. Phylogenetic analyses of the glycoprotein nt and aa sequences were carried out using Mega11 [25]. Maximum likelihood trees were built using the General Time Reversible model with the Gamma Distributed (G) pattern for the nt analysis while the Jones–Taylor–Thornton (JTT) with the G algorithm model was applied for the aa sequence analysis. Validity of the trees was tested using 2000 bootstraps for each.

### 2.3. Glycoprotein Sequence Analysis and 3D Structure Prediction

Primary, secondary and tertiary structures for the predicted glycoprotein of each LNYV subgroup were analysed and compared for any subgroup-specific features. Structures were predicted for the LNYV G sequences of isolates AU2 (SI from Australia, NC_007642 [16]), HV33 (SI from NZ, ON799188, the N gene previously published as NZ6 [17]), HV19 (SII, ON799187, the N gene previously published as NZ1 [26]) and an in silico variant form of NZ1 (DIII mutant) using the published 3D structure of vesicular stomatitis Indiana virus (VSV, YP009505325) for comparison [27]. Tertiary structures were predicted for aa 26–503 using I-TASSER (https://zhanggroup.org/I-TASSER/, accessed on 2 October 2020) [28,29] and Robetta (robetta.bakerlab.org accessed on 6 February 2021) [30] with the default settings, with topologies taken from UniProt (www.uniprot.org, accessed on 22 February 2021) [31]. Structures were validated using ERRAT, Verify3D and PROCHECK within the SAVES platform (saves.mbi.ucla.edu, accessed on 7 February 2021) [32,33,34]. Structures with the best scores (an ERRAT value above 50%, a Verify3D value above 80% and Ramachandran Plot above 90% [35,36] were chosen as the final structures and analysed using PDBsum (http://www.ebi.ac.uk/thornton-srv/databases/cgi-bin/pdbsum/, accessed on 4 February 2021). Structural domains were compared using PyMOL (PyMOL Molecular Graphics System. DeLano Scientific LLC, San Carlos, CA, USA. http://www.pymol.org, accessed on 2 October 2020), with the organisational structure of the LNYV glycoprotein being based upon that of Roche et al. [37]. Potential glycosylation sites, those with a value of more than 0.5, were identified using NetNGlyc 1.0 (http://www.cbs.dtu.dk/srvices/NetNGlyc/, accessed on 19 January 2021) [38].

## 3. Results

### 3.1. Sequence Analysis of the Glycoprotein Nucleotide and Amino Acid Sequences

Phylogenetic analysis of the G nt and aa sequences correlated with previous analysis of the nucleocapsid sequences [17], showing that the LNYV population in New Zealand is split into subgroup I and II (Figure 1B,C). This has also been observed previously for the Australian population [16]. No correlation was observed with sampling location within New Zealand, although no SII samples were identified from Levin in the south of the North Island. The Australian SI sequence shares an ancestor within the New Zealand SI sequences but is distinct, suggesting a country-based difference, as seen for the nucleocapsid sequences [17].

Comparison of the LNYV SI and SII glycoprotein sequences with the previously published LNYV SI sequence, LNYV-AU [22] and LYMoV [23], showed that this region of the genome remains highly conserved between LNYV isolates from New Zealand and Australia (Table 1). The New Zealand SI isolates showed 93.8–95% and 97.3–98.4% sequence similarity with LNYV-SI AU at the nt and aa levels, respectively. The SII isolates showed around 83% and 93.6–94.2% nt and aa similarity, respectively. While the SI isolates were more similar to the LNYV-AU SI isolate, both SI and SII isolates were more similar to the Australian sequence than they were to the LYMoV sequence, as expected.

### 3.2. Glycoprotein Primary, Secondary and Tertiary Structure Analysis

To investigate possible explanations for the more rapid dispersal of SII throughout Australia, and potentially through New Zealand, a 3D structure for the post-fusion form of each LNYV G protein sequence was predicted using the VSV G protein ectodomain structure as the model [37]. Validation analyses demonstrated the structures predicted by Robetta were more reliable than those produced by I-TASSER (Appendix A). Thus, only the structures using Robetta are shown. Figure 2A shows the overall topology of the LNYV-AU glycoprotein to be the same as that of the VSV glycoprotein, with different lengths of each region, except for the transmembrane region, reflecting the larger size of the LNYV protein. Figure 2B shows the overall domain structure between VSV and LNYV-SI from Australia to be very similar also, with the greatest difference in domain lengths at the C termini. High identity between the LNYV G aa sequences indicates they would all have the same topology and domain structures (Appendix A). The 3D arrangement of each domain is similar between each LNYV structure and between LNYV and VSV (Figure 2C). Adjustments made to accommodate the extra length of the LNYV ectodomain (476 aa compared to 406 for VSV) resulted in some domains being elongated, particularly Domains I (DI) and DIV.

#### 3.2.1. Domain I

DI is made from two regions within the glycoprotein and comprises about 116 amino acids in LNYV and about 90 in VSV. Several β-sheets are present in this domain in all structures, but the number and arrangement are inconsistent between the LNYV structures and between the structures of LNYV and VSV (Figure 2B,C and Figure 3A,B). LNYV-SII has the least number of β-sheets; secondary structure analysis indicated that LNYV-SII has no β-sheet between amino acids 310–329 (Figure 3B). In contrast, LNYV-SI from both Australia and New Zealand has two β-sheets in this region. Other notable aa differences are indicated in Figure 3B; for example, aa position 330 of LNYV-SII is a serine while it is arginine for LNYV-SI from both Australia and NZ. This may influence the length of the b-sheet in the position. Glycosylation was not predicted for LNYV at the N320 equivalent site as previously reported for VSV but was predicted at N3 [37]. However, the potential value for the N3 sites is just below the 0.5 threshold for likely glycosylation (Table 2) [38].

#### 3.2.2. Domain II

The comparison of Domain II (DII) structures is shown in Figure 2B,C and Figure 3C,D. This domain is formed from three regions of the primary sequence (Figure 2B). In all of the LNYV and VSV glycoprotein structures, DII consists of several α -helices. However, there are differences with regards to their number and arrangement (Figure 3C). VSV is predicted to have four, LNYV-SI from Australia and LNYV-SII have five, while LNYV -SI from New Zealand has six. Further, the arrangements of these helices were similar between the LNYV structures but dissimilar between LNYV and VSV. In LNYV, this domain has about 132 residues, with several non-conserved aa, such as at positions 411, 438, 442, 450 and 467 (Figure 3B). This domain in LNYV-SII appears to have two disulphide bonds. No glycosylation sites were predicted for this domain.

#### 3.2.3. Domain III

Domain III (DIII) of the LNYV glycoprotein ectodomain consists of 93 aa located in two segments in the structure (Figure 2B,C and Figure 3E,F). Based on the general appearance, there are structural similarities within this domain of LNYV and VSV; however, DIII of LNYV-SII shows the most distinct structure. Furthermore, a disulphide bond is predicted to connect cysteine (C)183 with C225 in LNYV-SII, but not in LNYV-SI from either Australia or New Zealand. Most other amino acid differences do not appear to influence the structure of this domain. Exceptions to this are positions 244 and 247 in the post-fusion protein structures (Figure 3F, aa positions 269 and 272 in the prefusion glycoprotein, Appendix A). At these positions, the amino acids were aspartate and alanine, respectively, in LNYV-SI from both Australia and New Zealand, while they were glutamate and serine in LNYV-SII. These amino acid changes appear to influence the formation of an a-helix in this region, with it being absent in LNYV-SII. In addition, all LNYVs are predicted to be glycosylated at N217 and N248 in DIII (N242 and N273 in the prefusion protein). Amino acid N217 had potential values of 0.45–0.52 while N248 had values of 0.55–0.61 (Table 2), suggesting a high probability of glycosylation of N217 for LNYV-SI from New Zealand, and of N248 for all LNYV subgroups [39].

#### 3.2.4. Domain IV

DIV shows high similarity between all structures (Figure 2B,C and Figure 3G,H). This domain comprises one region from the primary sequence and consists of 130 residues in LNYV, and amino acid differences did not appear to affect secondary structures. No glycosylation sites were predicted within this domain for any of the LNYV structures where there was one for VSV. Each 3D structure conserved hydrophobic amino acids at their base, such as W72, F74, Y120 and A121 (W97, F99, Y145 and A146 in the prefusion protein).

### 3.3. In Silico Mutagenesis of LNYV-SII G Protein

The correlation of amino acid changes within DIII at positions 244 and 247 with the absence of an alpha-helix in SII was of interest. These amino acids of LNYV-SII were changed to those of LNYV-SI, that is, from E244 and S247 to D244 to A247, and the impact on the predicted secondary and tertiary structures investigated (Figure 2B,C and Figure 3A–H). While these changes were made in DIII, the impact of these changes compared to LNYV-SII could be seen in the other domains as well (Figure 3A–H). For example, a predicted disulphide bond was lost in each domain (Figure 3B,D,F,H). In DI, **β** -sheets between positions 377–389 were elongated, while an α -helix around position 322 appeared, and one around position 358 was lost. In DIII, not only did an α -helix appear near the amino acid changes, but the α -helices around positions 35–47, 85–97, and 252–256 were also elongated, similar to LNYV-SI. A loop also appeared at position 198–202. In addition, support for glycosylation at each of the three predicted sites, N3, N217 and N248, remained the same, except for N248, which was reduced from 0.6 to 0.55. These findings suggest that these two amino acid positions are important in the formation of the functional LNYV glycoprotein.

## 4. Discussion

It was suggested that the rapid dispersal of LNYV-SII may have contributed to the apparent extinction of LNYV-SI in Australia and that this may have been due to more efficient transmission by the insect vector [17]. The glycoprotein of rhabdoviruses has a role in attaching the virion to their insect vectors; thus, the aim of our study was to investigate differences between LNYV-SI and -SII within the glycoprotein that might support this hypothesis and to explain the difference observed in the two populations. This study focused on LNYV from New Zealand due to the ready availability of samples; the hypotheses generated from this study should be tested with Australian samples to provide further supporting evidence.

Phylogenetic analysis of the G gene and protein expands the previous sequence information of LNYV within New Zealand [17] to other regions outside Auckland, in both the North and South Islands. The population structure of LNYV in New Zealand was confirmed as consisting of two subgroups—SI and SII—with the subgroup allocation based on the glycoprotein sequences correlating with the samples’ nucleocapsid sequences at both the nt and aa levels [18]. These SII sequences also appeared to be more recent than the SI sequences, as found previously with the nucleocapsid sequences [17]. The confirmation of such LNYV population structure using glycoprotein sequences suggests that the differences in dispersal of the LNYV subgroups could be brought about by differences in attachment efficiency to the aphid vector. Analysis of more glycoprotein sequences, particularly from Australia, will strengthen these findings.

The possible role of the glycoprotein in enabling the dispersal of LNYV-SII was investigated through the analysis of predicted 3D protein structures. The VSV glycoprotein is considered as a class III viral membrane fusion protein [40,41]. These proteins are known to catalyse the fusion of viral and cellular membranes as part of the entry process of enveloped viruses into their host cell [42]. Class III fusion proteins are characterised by trimerisation as part of their pre- and/or post- fusion conformation, with a coiled-coil region at the centre in its post-fusion form and a **β** -sheet-rich fusion domain at its N-terminus [43]. The predicted structures for the ectodomains of LNYV-SI and -SII presented were consistent with these features and showed high similarity to the structure of VSV glycoprotein. This suggests that LNYV glycoproteins may also function similarly to class III viral membrane fusion proteins. Further, the coiled structure of the DII domain serves as the trimerisation site of VSV (Sun et al., 2010). Hence, the consistent α -helical structure of this domain among all the predicted LNYV glycoproteins suggests that DII could have the same role for LNYV glycoproteins. Martin et al. [44] unexpectedly found that LNYV G proteins did not self-interact within *Nicotiana benthamiana* cells, suggesting the protein does not form trimers; however, this may have been due to constraints of the BiFC assay used or differences between plants and insect cells. Given that trimerisation appears to be a universal characteristic of glycoproteins involved in membrane fusion, it is likely that the LNYV G proteins also form trimers. The VSV glycoprotein binds the low-density lipoprotein receptor (LDL-R), allowing the virion to enter the cell, as in Rabies virus, via actin-dependent clathrin-mediated endocytosis [45,46]. The reduced pH of the endocytic vesicles causes the G protein to change its trimeric conformations from pre- to post-fusion, driving fusion of the virion and vesicle membranes [37,47]. This allows entry of the viral genome into the cell cytosol. While there is no direct evidence for LNYV glycoprotein having a role in viral entry into aphid cells, it is not unreasonable to assume that it has this role and the virus enters in a similar manner. It is also tempting to speculate that, based on its similar structure to VSV, LNYV G proteins bind LDL-R on the surface of aphid cells. Nikolic et al. [48] identified K47 and R354 of the VSV G protein as being essential for binding with LDL-R, but not for membrane fusion. These authors noted that these residues are not conserved and, therefore, LDL-R cannot be generalised as a receptor; thus, LDL-R is unlikely to be the receptor for LNYV as these residues are not conserved within this virus.

The amino acids E244 and S247 within DIII of the post-fusion protein may have a role in the rapid dispersal of SII in Australia and may explain the population differences observed between Australia and New Zealand [17]. In silico mutation of these two amino acids led to loss of a predicted disulphide bond from each domain, reducing the total number in LNYV-SII from the greatest to the least. Glycosylation sites were predicted at positions N3, N21 and N248 for LNYV-SI and -SII. Altering E244 and S247 changed DIII of LNYV-SII to be more like that of SI, which, in turn, reduced the potential for glycosylation of N248. The amino acids E244 and S247 within DIII of LNYV-SII may provide a combinatorial approach to enhance aphid transmission over that of LNYV-SI. G protein glycosylation is important for protein function, interaction with host cells and evading the host’s immune system [49,50,51]. Interestingly, major antigenic sites of VSV are within DI and DIII [52], the domains that showed the greatest differences between the different LNYV structures. By creating a G protein structure that is more stable and easier to be glycosylated, this may have enhanced the ability of LNYV-SII to evade the aphid’s immune system as well as its interaction with aphid host cells, allowing it to disperse more rapidly than LNYV-SI. This may explain how SII came to supplant SI in Australia, assuming SII in Australia is similar to that in New Zealand. In New Zealand, both LNYV-SI and SII are present, possibly because SI in New Zealand has two potential glycosylation sites (N217, N248), while SI in Australia and SII has only one. Yamada et al. [53] suggested that increasing the number of glycosylation sites could increase virus production in cells, which might have allowed LNYV-SI to remain present in New Zealand. Alternatively, the addition of the disulphide bond may have given the SII G protein a greater relative enhancement in stability in Australia than in NZ, as the higher temperatures in Australia are more likely to lead to protein unfolding. Rather than a more efficient transmission of SII being responsible, it may have been that the G protein of LNYV-SI in Australia was more likely to experience unfolding, making it unavailable to the aphid vector. The lower ambient temperatures in NZ may have provided an environment where the lower stability of the SI glycoprotein was not a disadvantage, allowing it to be maintained in the NZ LNYV population. The next logical step would be to create recombinant viruses with these mutations to test for their effect on biological activity of the G protein. However, this is difficult under the legal framework of New Zealand; we seek collaborations to test our hypotheses.

RNA viruses, such as LNYV, exist as quasi-species, with the emergence of new strains and species being a continuous process. Phylogenetic analysis showed that LNYV-SII emerged after SI, likely because of sequence differences that arose through replication that provided some advantage to allow it to disperse rapidly and become fixed in the population. This study suggests that specific changes that contributed to this were D244E and A247S. Expanding this study to include Australian SII sequences, as well as testing the effect of these amino acid changes on glycoprotein stability and its membrane fusion mechanism, will be essential to confirm this finding.

## Figures and Tables

**Figure 1 viruses-14-01574-f001:**
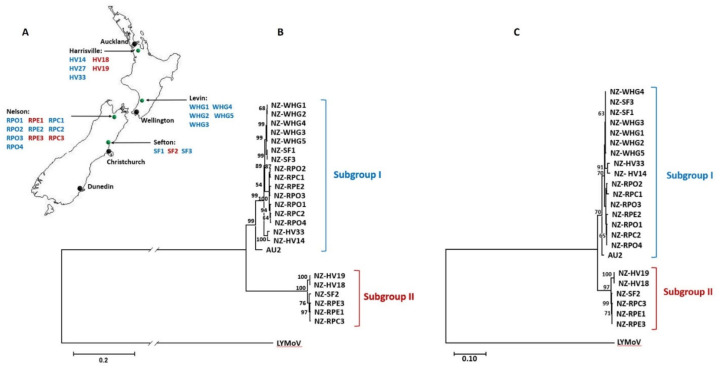
LNYV SI and SI samples collected from New Zealand sites. (**A**) Map of NZ illustrating the locations of samples. Black pins indicate major NZ cities, and green pins indicate sample sites. Blue text represents isolates that were diagnosed as LNYV subgroup I and red text represents isolates that were diagnosed as LNYV subgroup II. Modified with permission from Darling [18]. (**B**,**C**) Maximum likelihood analysis of glycoprotein nt and aa sequences, respectively, with LYMoV as the outgroup. Bootstrap values greater than 50 are indicated at the branch nodes. The scales indicate the number of substitutions per site.

**Figure 2 viruses-14-01574-f002:**
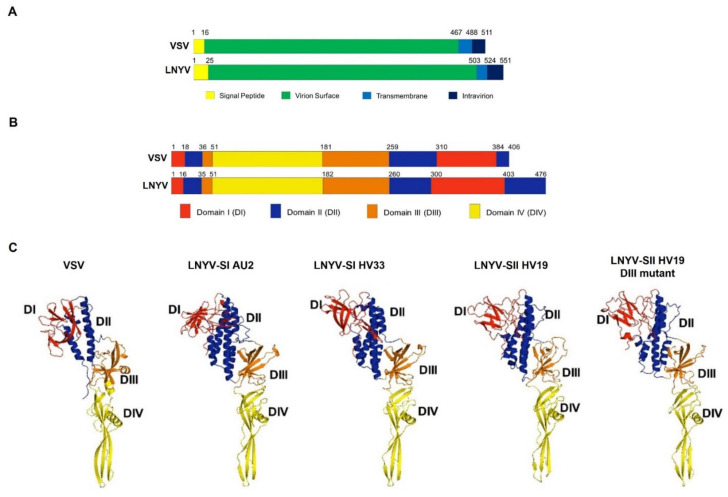
LNYV SI and SI samples collected from New Zealand sites. (**A**) Comparison of the topology of the pre-fusion forms of LNYV and VSV glycoproteins. (**B**) Comparison of the domain structure of the post-fusion forms of LNYV and VSV glycoproteins. (**C**) Ribbon models of the predicted 3D structures for the post-fusion glycoproteins of VSV, LNYV subgroups I (HV33) and SII (HV19) as well as the in silico DIII mutant of HV19. Domains are coloured according to that shown in (**B**).

**Figure 3 viruses-14-01574-f003:**
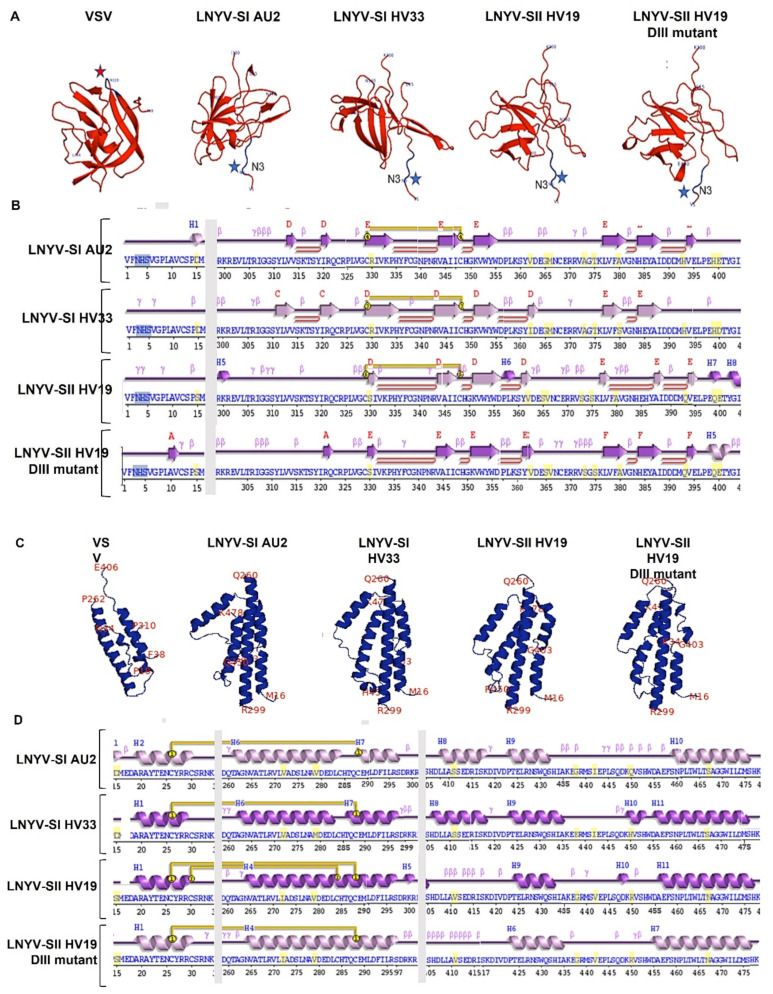
Tertiary and secondary structures of each domain of each post-fusion glycoprotein. (**A**,**C**,**E**,**G**) Predicted 3D structures for Domain I (DI), Domain II (DII), Domain III (DIII) and Domain IV (DIV), respectively. (**B**,**D**,**F**,**H**) Alignments of the predicted secondary structures for DI, DII, DIII and DIV, respectively. (**B**) Alignment of DI aa positions 1–16 and 300–405; (**D**) alignment of DII aa positions 15–35, 259–304 and 405–478; (**F**) alignment of DIII aa positions 34–51 and 180–261; (**H**) alignment of DIV aa positions DIV, 49–181. Amino acids that are not conserved are marked in yellow. The low-possibility glycosylation sites are marked in blue in the alignment and blue stars in the 3D structure. The high-potential glycosylation sites are marked in red in the alignment and red stars in the 3D structure. Disulphide bonds are marked by yellow lines and loops by red lines. The altered amino acids are marked in purple.

**Table 1 viruses-14-01574-t001:** Nucleotide and amino acid sequence similarities between the glycoprotein sequences of each LNYV isolate and the G protein from LNYV-SI AU (NC_007642) and LYMoV (EF687738).

LNYV Subgroup	Isolate Name	Accession Number	Similarity with LNYV-SI AU2 (%)		Similarity with LYMoV (%)	
			nt	aa	nt	aa
SI	SF1	ON799199	94.8	98.4	56.5	49.8
	SF3	ON799201	94.8	98.4	56.5	49.8
	WHG1	ON799202	94.9	98.4	56.4	49.8
	WHG2	ON799203	94.9	98.4	56.4	49.8
	WHG3	ON799204	95	98.4	56.5	49.8
	WHG4	ON799205	95	98.4	56.5	49.8
	WHG5	ON799206	94.9	98.2	56.5	49.8
	RPO1	ON799195	93.9	97.8	56.5	50
	RPO2	ON799196	94.2	97.7	56.5	49.8
	RPO3	ON799197	94.2	98.2	56.6	50.2
	RPO4	ON799198	94.1	98	56.3	50
	RPE2	ON799193	93.8	97.8	56.5	50.2
	RPC1	ON799189	93.9	97.5	56.6	50.4
	RPC2	ON799190	94.1	98	56.4	50
	HV14	ON799185	94	97.3	56.2	49.6
SII	SF2	ON799200	83.2	94.2	55.3	49.8
	RPE1	ON799192	83	94.2	55.2	49.8
	RPE3	ON799194	83	94.2	55.3	49.8
	RPC3	ON799191	83	94	55.3	49.8
	HV18	ON799186	83	93.6	55	49.6

**Table 2 viruses-14-01574-t002:** Prediction of N-glycosylation sites for each LNYV structure.

Sample	aa Position	Sequence	Potential	Likely Glycosylation
LNYV-SI AU2	3	NHSV	0.4867	-
	217	NSTT	0.4546	-
	248	NDSK	0.5548	+
LNYV-SI HV33 (NZ6)	3	NHSV	0.4863	-
	217	NSTT	0.521	+
	248	NDSK	0.5546	+
LNYV-SII HV19 (NZ1)	3	NHSV	0.4864	-
	217	NSTT	0.4656	-
	248	NDSK	0.6124	+
LNYV-SII HV19 (NZ1) DIII mutant (E244D, S247A)	3	NHSV	0.4863	-
	217	NSTT	0.4654	-
	248	NDSK	0.5547	+

## Data Availability

Data are contained within the article or Appendix A.

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
