# Peer review of "Is the Glycoprotein Responsible for the Differences in Dispersal Rates between Lettuce Necrotic Yellows Virus Subgroups?"

_viruses, 2022, doi:10.3390/v14071574_

Round 1
Reviewer 1 Report
The manuscript is well written and the topic is very sounded, just a few comments we need the authors to answer.
Materials and Methods
“Symptomatic lettuce leaves were collected from various locations in Aotearoa New
Zealand (Figure 1A). Isolates HV14 and HV18 were provided by Colleen Higgins (Auck land University of Technology [AUT], NZ), while the remaining isolates were provided by John Fletcher (The Institute for Plant and Food Research, NZ).”
- The questions here…. Symptomatic lettuce leaves …you mean infected…So, by which kind of virus isolate?
- “the remaining isolates were provided by John Fletcher”…..which one…please included here?
Author Response
Hello,
Please see the attachment.
Thank you very much.

Reviewer 2 Report
Prabowo and colleagues in this study did the detailed bioinformatic analysis for the role of glycoprotein in dispersal rates. Critical aa sites/differences have been identified for their potential function in the LNYV subgroups of SI and SII. One major concern is the experimental evidence is missing. I strongly recommend the authors to construct the reverse genetics system of LNYV, and introduce the critical aa mutations into viral genome to confirm the biological function.
Author Response
Hello,
Please see the attachment.
With great thanks.

Round 2
Reviewer 2 Report
it is reasonable.